# Leveraging Inter-Layer Dependency for Post -Training Quantization

**Changbao Wang**
Ant Technology Group Co., Ltd.
changbao.wcb@antgroup.com

**Dandan Zheng**
Ant Technology Group Co., Ltd.
yuandan.zdd@antgroup.com

**Yuanliu Liu**
Ant Technology Group Co., Ltd.
yuanliu.lyl@antgroup.com

**Liang Li**
Ant Technology Group Co., Ltd.
double.ll@antgroup.com

## Abstract

Prior works on Post-training Quantization (PTQ) typically separate a neural network into sub-nets and quantize them sequentially. This process pays little attention to the dependency across the sub-nets, hence is less optimal. In this paper, we propose a novel Network-Wise Quantization (NWQ) approach to fully leveraging inter-layer dependency. NWQ faces a larger scale combinatorial optimization problem of discrete variables than in previous works, which raises two major challenges: over-fitting and discrete optimization problem. NWQ alleviates over fitting via a Activation Regularization (AR) technique, which better controls the activation distribution. To optimize discrete variables, NWQ introduces Annealing Softmax (ASoftmax) and Annealing Mixup (AMixup) to progressively transition quantized weights and activations from continuity to discretization, respectively. Extensive experiments demonstrates that NWQ outperforms prior state-of-the-art approaches by a large margin: 20.24% for the challenging configuration of MobileNetV2 with 2 bits on ImageNet, pushing extremely low-bit PTQ from feasibility to usability. In addition, NWQ is able to achieve competitive or better results with only 10% computation cost of previous works.

## 1  Introduction

Deep learning has seen a boom of architectures with ever-increasing capabilities and capacity. Huge capacity, however, always comes with boosting parameters, resulting in large computation cost and memory footprint. Researchers have utilized strategies like neural architecture search (NAS) [4, 11, 29, 49], network pruning [12, 19], and knowledge distillation (KD) [18, 21] to design compact networks and improve their accuracy. Furthermore, neural network quantization [7, 15, 22, 25, 28] is typically applied along with the methods mentioned above to reduce the computation cost and memory footprint. Quantization represents neural networks with lower bit precision, usually 8 bits in practice, and saves 75% of memory while offering $2 \sim 4\times$ speedup[26].

Network quantization techniques are typically categorized into Quantization Aware Training (QAT) and Post-training Quantization(PTQ) based on the dependency on labeled data and the necessity of weight tuning. Generally, PTQ techniques are preferred in industry because they rely less on labeled data and take into account privacy concerns as well as compute resources. This paper focuses on PTQ.

PTQ usually suffers from significant accuracy degradation, especially for low-bit quantization which offers much more memory reduction and speedup. To improve the accuracy of PTQ, much effort

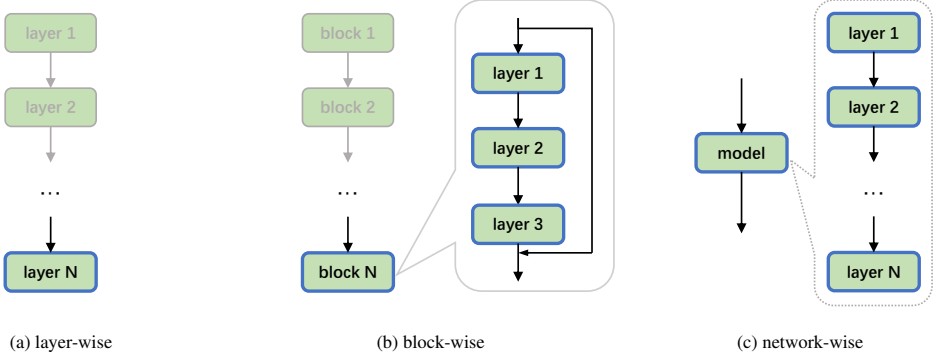

|          |          |          |
|----------|----------|----------|
| (a) layer-wise | (b) block-wise | (c) network-wise |

Figure 1: **layer-wise vs. block-wise vs. network-wise**. Gray nodes are frozen while highlighted nodes are learnable (quantization parameters only). The term *block* here denotes residual unit in modern CNN (*e.g.* Residual Bottleneck for ResNets).

has been devoted [5, 32, 34, 35, 39]. AdaRound[32] has recently discovered that round-to-nearest appears to be optimal for single weight but not for the entire layer. The rounding errors of weights in a single layer have the capability of compensating for each other, contributing to a smaller layer-wise quantization error. We term this property as *intra-layer dependency*. Motivated by this property, AdaRound[32] proposes to learn the round policy layer by layer to improve accuracy.

Inspired by AdaRound[32], we hypothesize that the quantization errors of all layers in a network are able to compensate for each other, resulting in a reduced quantization error. We refer to this capability as *inter-layer dependency*. However, AdaRound[32] assumes that layers are mutual-independent and derives a layer-diagonal Hessian, leading to a layer-wise solution(Figure 1a). BRECQ[27] points out that the dependency should not be ignored and the assumption of AdaRound[32] is not accurate, they instead utilize the inter-layer dependency within each block to build a block-wise approach(Figure 1b). However, BRECQ[27] assumes that blocks are mutual-independent. We argue that neither the approximations of AdaRound nor that of BRECQ are accurate enough, the inter-layer dependency should be leveraged in a network-wise manner. To achieve this, we propose a Network-Wise Quantization (NWQ, Figure 1c) scheme to fully leverage the inter-layer dependency. AdaRound and BRECQ separate a network into sub-nets and train them sequentially, this process makes the preceding layers unaware of succeeding layers. NWQ solves this problem via training the entire network end-to-end, where layers are aware of each other and they are able to cooperate for a more optimal solution. However, quantizing the entire network end-to-end is a combinatorial optimization problem where huge number of discrete parameters from all layers are combined together for optimization. This optimization problem will raise two major challenges that result in significant accuracy degradation.

The first challenge is over-fitting. Extending the reconstruction unit from layer/block to entire network will increase parameters dramatically and thus raises the risk of over-fitting over a typically small calibration set. To alleviate this issue, NWQ applies Activation Regularization (AR), a variant of L2 regularization, on intermediate representation to keep the quantized activations as close to the floating-point counterpart as possible. The benefits of AR are two folds. First, AR regularizes quantized activation flow to prevent over-fitting. Second, it works as a quantization loss function to encourage minimal intermediate quantization error. AR turns out to be both simple and effective.

Another challenge is discrete optimization. Based on AdaRound, NWQ learns the weight round policy and activation quantization parameters, both of which involve in discrete variables. However, the substantially increased depth and discrete variables make it more difficult to converge compared to layer-wise[23, 32] and block-wise[27, 40] approach. The insufficient convergence will leave the discrete variables (weight round policy) in a state of continuity rather than discretization, leading to great performance drop. To solve this problem, we propose Annealing Sofmtax (ASoftmax) and Annealing Mixup (AMixup) to better optimize discrete weights and activations. Specifically, ASoftmax provides a mechanism to guarantee the discritization of round policy learning via annealing temperature. AMixup resolves the train-test inconsistency problem indtroduced by QDROP[40] via gradually decreasing the floating-point percentage in mixed activations.

Equipped with AR, ASoftmax and AMixup, we are able to build our NWQ framework that fully leverages *inter-layer dependency* to achieve higher accuracy than existing works.

To summarize, this paper makes the following contributions:

1. We propose a novel network-wise quantization framework for PTQ to fully leverage *inter-layer dependency*.

2. We analyze the combinatorial optimization problem introduced by network-wise quantization and propose AR, ASoftmax and AMixup to solve the over-fitting and discrete optimization problem.

3. We establish a new state-of-the-art through significantly improving PTQ by a large margin: 20.24% for MobileNetV2 with 2 bits and thus push the limit of extremely low-bit PTQ from feasibility to usability.

## 2   Related Work

Network quantization is commonly divided into QAT [7, 15, 22, 25, 48] and PTQ [27, 32, 33, 47]. QAT methods tend to minimize task error by training weights and quantization parameters jointly with the entire labeled dataset and thus require a large number of computing resources, whereas PTQ methods focus on finding optimal quantization parameters for off-the-shelf well-trained networks with much less data and compute resources than QAT. PTQ can further be classified into *statistic-based PTQ* and *learning-based PTQ*. Statistic-based PTQ methods [33, 35, 47] first estimate the distribution of activation on calibration data, and then find the optimal quantization parameters that minimize local quantization errors heuristically. Learning-based PTQ methods[23, 27, 32, 40] dedicate to slightly weight tuning and quantization parameters learning. Our work follows the learning-based PTQ scheme.

**Learning-based PTQ**: AdaRound[32] adopts the regime of learning the optimal round policy layer by layer. BRECQ [27] succeeds in extending AdaRound [32] to a block-wise reconstruction and concludes that block-wise quantization is superior to network-wise method. We argure that both layer-wise and block-wise approach is sub-optimal because the preceding layers/blocks are unaware of succeeding layers/block, thus the dependency they leveraged is unidirectional. Some works focus on the sequence of weight and activation quantization. AdaQuant [23] proposes to tune weights and activation quantization parameters separately for each layer to reduce over-fitting, while QDROP [40] finds that incorporating activation quantization into weight tuning can achieve higher accuracy.

**Activiation Regularization**: Large values in the learned dense representation can be a sign of being overfit. To encourage small activations, [31] employs L2 regularization on hidden state of recurrent neural networks (RNN), while various normalizations [2, 24, 38, 42] focus on centralize the activation distribution to regularize large values. Some prior works attempt to learn sparse representation. Dropout and its variants [1, 10, 36, 37, 41] randomly drop activations from the neural network to encourage sparsity. ReLUs[16, 20, 30, 43] dedicate to learning sparse representation via rectifying negative values. These techniques aim to regularize activations when training a neural network from scratch, which inapplicable for PTQ where the networks are off-the-shelf and well-trained, thus the activation distributions are deterministic and immutable. Instead of encouraging small and sparse activations, NWQ apply activation regularization on intermediate representation to encourage small offsets between quantized activations and the floating-point counterpart. AdaRound[32] and BRECQ [27] also employ quantization losses on intermediate activations but those losses works as quantization objectives rather than regularization, while the quantization objective of NWQ is to minimize the quantization error of the final network output and the regularization on intermediate activations is to prevent optimization process from over-fitting.

**Discrete Optimization for Quantization**: Network quantization optimizes discrete variables via gradient backpropagation. However, the quantization process introduces a round function which is indifferentiable. Some works on QAT focus on approximating the gradient through the round function [3, 6, 13, 14, 17, 44]. For PTQ, AdaRound [32] introduces additional discrete parameters(the binary round policy) and optimize them via Rectified Sigmoid with an explicit Regularization (RSeR) loss function. However, RSeR is adversarial to quantization loss and thus difficult to convergence in network-wise quantization. QDROP [40] proposes to randomly drop quantized activations to improve

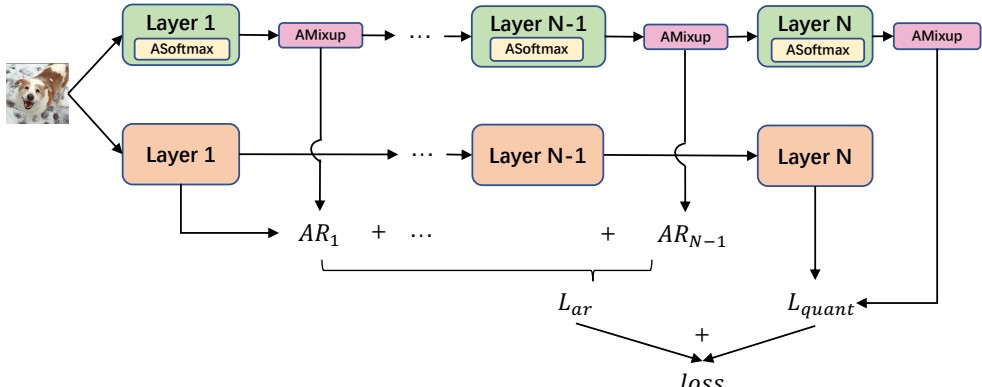

Figure 2: **Overfiew of NWQ**. We use L2 loss for both $AR_i$ and $L_{quant}$. The green is simulated quantization network and the orange is the floating-point counterpart. We freeze all weights and train weight round policy and activation quantization parameters only.

the flatness of optimization landscape, but introcues train-test inconsistency problem which results in performance degradation if used in network-wise PTQ.

## 3 Approach

Network-Wise Quantization (NWQ) is a simple end-to-end post-training quantization approach. The structure of NWQ is illustrated in Figure 2. The optimization objective is composed of output quantization loss and layer-wise AR loss. Meanwhile, the annealing approaches (Asoftmax and AMixup) are introduced in training process, which enable the quantized weights and activations to gently converge from continuous state to discrete state, lowering training complexity and improving accuracy.

### 3.1 Activation Regularization

NWQ leverages Activation Regularization (AR) to penalize quantized intermediate activations that are substantially away from the floating-point counterparts, thereby encouraging the activations to remain unchanged.

Formally, we define AR by:

$$L_{ar}(\overline{G}, G) = \sum_{l=1}^{N-1} \|\overline{a}_l - a_l\|_F^2, \tag{1}$$

while $G$ represents floating-point network and $\overline{G}$ is the simulated quantized counterpart. $a_l$ and $\overline{a}_l$ denote the floating-point and the simulated quantized activation of the $l$-th layer, respectively. $N$ represents the number of total layers and $\|\cdot\|_F$ denotes the Frobenius norm.

Our network quantization error $L_{quant}(\overline{a}_N, a_N)$ is defined by:

$$L_{quant}(\overline{G}, G) = \|\overline{a}_N - a_N\|_F^2,$$

then the final objective can be formulated as:

$$L(\overline{G}, G) = L_{ar} + L_{quant} = \sum_{l=1}^{N} \|\overline{a}_l - a_l\|_F^2. \tag{2}$$

Practically, $L_{ar}$ is not restricted to be utilized layer by layer, it can also be performed block by block and stage by stage, as BRECQ [27] defined. We will discuss how the granularity affects the performance in Sec. 4.2.1. By default, we employ layer-wise AR in our experiments.

## 3.2 Annealing Softmax

For a weight vector $\mathbf{w}$, AdaRound [32] proposes to learn the round policy by:

$$\hat{\mathbf{w}}_i = clip\left(\lfloor\frac{\mathbf{w}_i}{s}\rfloor + h(\mathbf{v}_i), q_-, q_+\right), \tag{3}$$

where $\lfloor\cdot\rfloor$ gives as output of the integral part and $clip(\cdot)$ clamps all values between $q_-$ and $q_+$. $\hat{\mathbf{w}}_i$ denotes the quantized weight for the $i$-th element of $\mathbf{w}$. We will omit $i$ below for simplicity. $s$ is the quantizer step size. $\mathbf{v}$ is a learnable continuous vector *w.r.t.* $\mathbf{w}$ and $h(\mathbf{v})$ is *recitified sigmoid* function (see Eqn.(4)) which is regularized to converge to $\{0, 1\}$.

$$h(\mathbf{v}) = clip(\frac{1.2}{1 + exp(-\mathbf{v})} - 0.1, 0, 1). \tag{4}$$

Based on Eqn.(4), AdaRound proposes Rectified Sigmoid with an explicit Regularization (RSeR) loss to discrete $h(\mathbf{v})$:

$$L_{RSeR}(h(\mathbf{v})) = 1 - |2h(\mathbf{v}) - 1|^\beta, \tag{5}$$

However, there are two limitations of RSeR. First, RSeR works only for binary round policy. Second, RSeR is adversarial to quantization loss. For example, for a linear transformation network, the quantization loss is

$$\begin{aligned} L_q &= \|\mathbf{w}^\intercal\mathbf{x} - s\hat{\mathbf{w}}^\intercal\mathbf{x}\|_F^2 \\ &= s^2\|(\frac{\mathbf{w}}{s} - \lfloor\frac{\mathbf{w}}{s}\rfloor - h(\mathbf{v}))^\intercal\mathbf{x}\|_F^2, \end{aligned} \tag{6}$$

where $\mathbf{x}$ denotes the input and we ignore the *clip* function in Eqn. (3).

Eqn. (6) indicates that quantization loss will prevent $L_{RSeR}$ from convergence (*i.e.*, $L_{RSeR}$ encourages $h(\mathbf{v})$ to converge to 0 or 1 while $L_q$ prefers $h(\mathbf{v})$ to remain $\frac{\mathbf{w}}{s} - \lfloor\frac{\mathbf{w}}{s}\rfloor$). Besides, this adversarial become more severe when extending layer/block-wise methods to network-wise quantization(see Sec. 4.2.2 and Table 4).

### 3.2.1 Formulation

To resolve the two problems of RSeR, we propose Annealing Softmax (ASoftmax). First, we extend the discrete optimization space of $h(\cdot)$ from $\{0, 1\}$ to $\mathcal{K}$, where $\mathcal{K}$ is a integer range and defined by:

$$\mathcal{K} = \{n, n+1, ..., m-1, m\}, n \in \mathbb{Z}_{\leq 0}, m \in \mathbb{Z}_{>0}, \tag{7}$$

where $n$ and $m$ are hyper-parameters that specify the discrete optimization space. Then, ASoftmax can be formulated as:

$$\hat{\mathbf{w}}^t = clip\left(\lfloor\frac{\mathbf{w}}{s}\rfloor + h^t(\mathbf{V}), q_-, q_+\right), \tag{8}$$

where $\mathbf{V}$ is a continuous learnable matrix and represents the logits of probability over $\mathcal{K}$. $h^t(\mathbf{V})$ is defined by:

$$h^t(\mathbf{V}) = \mathbb{E}_{k\sim p(k|\mathbf{V}, \tau^t)}\left[k\right], \tag{9}$$

$$p(k = \mathcal{K}_i|\mathbf{V}, \tau^t) \propto exp(\mathbf{V}_i/\tau^t), 0 \leq i < m - n + 1, \tag{10}$$

where $t$ represents for $t$-th iteration. $\tau^t$ is a temperature that decays linearly to transition $h^t(\mathbf{V})$ from continuity to discretization. For example, when $\tau^t$ is large, $p(k|\mathbf{V}, \tau^t)$ is a continuous vector of probability and $h^t(\mathbf{V})$ is the *expectation* of the discretization. When $\tau^t$ turns tiny, $p(k|\mathbf{V}, \tau^t)$ becomes a one-hot vector and $h^t(\mathbf{V})$ selects $k$ by $p(k|\mathbf{V}, \tau^t) = 1$. $\tau^t$ is simply decayed by:

$$\tau^t = \frac{(\tau^T - \tau^0) \cdot t}{T} + \tau^0, \tag{11}$$

where $\tau^0$ and $\tau^T$ are hyper-parameters of initial and final temperature.

ASoftmax is a general formation of learning wider range of discrete parameters than binary. It easily degenerates to round policy learning by setting $n = 0$ and $m = 1$. We use $\tau^T = 0.01$ and $\tau^0 = 1$ by default.

### 3.2.2 Initialization

According to our observations, initialization has significant impacts on learning round policy. Given the weight quantizer step size $s$, which has been computed using statistic-based PTQ methods, we attempt to initialize $\mathbf{V}$ in order to make $s \cdot \hat{\mathbf{w}}^0$ as close to floating-point $\mathbf{w}$ as possible at the start. To achieve this goal, we first calculate the un-normalized probability by:

$$\sigma'(\mathbf{V})_i = \frac{1}{|\mathcal{K}_i - (\frac{w}{s} - \lfloor \frac{w}{s} \rfloor)|}, 0 \leq i < m - n + 1. \tag{12}$$

Then, we normalize the probability:

$$\sigma(\mathbf{V})_i = \frac{\sigma'(\mathbf{V})_i}{\sum_{k=0}^{m-n} \sigma'(\mathbf{V})_k}, \tag{13}$$

Finally, the initialization for $\mathbf{V}_i$ is $log(\sigma(\mathbf{V})_i)$. It can be easily proved (see Appendix A) that the initialized $s \cdot \hat{\mathbf{w}}^0$ equals $\mathbf{w}$ as long as the integer range $\mathcal{K}$ is symmetric *w.r.t.* 0.5 (*i.e.* $n = 1 - m$).

### 3.3 Annealing Mixup

NWQ considers activation quantization as a discrete optimization problem and solves it via continuous relaxation. Different from ASoftmax which is applied on element-wise level, AMixup works on tensor-wise level: we randomly mix up the floating-point activations with the simulated quantized activations. To encourage continuous relaxation converge to discretization, NWQ progressively decreases the percentage of floating-point activations, so the mixed activations will become fully quantized activation gradually.

Formally, the percentage of floating-point activations is defined by:

$$P(t) = \frac{(P_e - P_s) \cdot t}{T} + P_s, \tag{14}$$

where $P(t)$ is the percentage of floating-point activation in the mixed activation for the $t$-th iteration. $P_s$ and $P_e$ denote the percentage at start and end respectively, and $T$ is the number of iterations. We use $P_s = 0.5$ and $P_e = 0$ by default.

AMixup is similar to QDROP [40] in the mixup operation. The key difference, however, is that AMixup introduces an annealing process. This is very important for NWQ. QDROP trains with mixed activations while testing with fully quantized activations, thus introducing inconsistency bias. This bias accmulates as network depth increases and leads to great accuracy degradation in network-wise quantization. AMixup is able to eliminate the inconsistency bias via the annealing process.

## 4 Experiments

In this section, we conduct extensive experiments with various architectures to verify our NWQ. We first compare our method with previous state-of-the-art, and then we delve into our important designs and fully explore the scalability, effectiveness, and efficiency of our solution. We report top-1 classification accuracy on the ImageNet[9] validation set.

**Implementation Details.** Our code is based on open-source codes BRECQ and QDrop. We apply layer-wise AR. For ASoftmax, we set $n = 0, m = 1$ and $\tau$ is decayed from 1 to 0.01. For AMixup, we set default $P_s$ as 0.5 and $P_e$ as 0. We randomly sample 1024 images from ImageNet train set and employ Cutmix [45] and Mixup[46] as data augmentation. The learning rates are 0.01 for round policy and 0.0004 for activation quantizer step size. We train for 20000 iterations with a mini-batch size of 32 in 8 Tesla V100 GPUs, taking $\sim$ 30 minutes for ResNet18, which is on par with BRECQ and QDROP. Our experiments are conducted on 5 architectures, including ResNet18(Res18), MobileNetV2(MNV2), RegNet-600MF(Reg600MF), RegNet-3.2GF(Reg3.2GF) and MnasNet(Mnas). Other settings remain same as QDROP[40] if not specified.

### 4.1 Comprehensive Comparison

Table 1 compares our NWQ with several main-stream PTQ approaches. Overall, NWQ outperforms previous works across all neural network architectures and bit widths, and our performance improves even more as the bit width decreases.

| | Bits(W/A) | Res18 | MobileNetV2 | Reg600MF | Reg3.2GF | MnasNet2.0 |
|---|---|---|---|---|---|---|
| Full Prec | 32/32 | 71.06 | 72.49 | 73.71 | 78.36 | 76.68 |
| ACIQ-Mix* [35] | 4/4 | 67.00 | - | - | - | - |
| ZeroQ* [5] | 4/4 | 21.71 | 26.24 | 28.54 | 12.24 | 3.89 |
| LAPQ* [34] | 4/4 | 60.30 | 49.70 | 57.71 | 55.89 | 65.32 |
| AdaQuant* [23] | 4/4 | 69.60 | 47.16 | - | - | - |
| Bit-Split* [39] | 4/4 | 67.56 | - | - | - | - |
| AdaRound* [32] | 4/4 | 67.96 | 61.52 | 68.20 | 73.85 | 68.86 |
| BRECQ* [27] | 4/4 | 69.60 | 66.57 | 68.33 | 74.21 | 73.56 |
| QDROP* [40] | 4/4 | 69.62 | 68.84 | 71.18 | 76.66 | 73.71 |
| QDROP† [40] | 4/4 | 69.88 | 69.15 | 71.45 | 77.02 | 74.09 |
| NWQ(Ours) | 4/4 | 69.85±0.04 | 69.14±0.12 | 71.92±0.07 | 77.40±0.04 | 74.60±0.09 |
| NWQ†(Ours) | 4/4 | **69.95**±0.07 | **69.60**±0.03 | **72.03**±0.09 | **77.42**±0.03 | **74.86**±0.03 |
| AdaQuant* [23] | 3/3 | 60.09 | 2.23 | - | - | - |
| AdaRound* [32] | 3/3 | 64.66 | 15.20 | 51.01 | 56.79 | 47.89 |
| BRECQ* [27] | 3/3 | 65.87 | 23.41 | 55.16 | 57.12 | 49.78 |
| QDROP* [40] | 3/3 | 66.75 | 57.98 | 65.54 | 72.51 | 66.81 |
| QDROP† [40] | 3/3 | 67.51 | 58.54 | 66.75 | 73.48 | 68.28 |
| NWQ(Ours) | 3/3 | 67.58±0.14 | 61.24±0.21 | 67.38±0.13 | 74.79±0.09 | 68.85±0.18 |
| NWQ†(Ours) | 3/3 | **68.15**±0.10 | **62.53**±0.05 | **68.31**±0.09 | **75.44**±0.08 | **70.77**±0.10 |
| BRECQ* [27] | 2/2 | 42.54 | 0.24 | 3.58 | 3.62 | 0.61 |
| QDROP* [40] | 2/2 | 54.72 | 13.05 | 41.47 | 55.11 | 28.77 |
| QDROP† [40] | 2/2 | 57.84 | 14.89 | 45.93 | 57.98 | 34.19 |
| NWQ(Ours) | 2/2 | 59.14±0.05 | 26.42±1.03 | 48.49±0.03 | 62.85±0.13 | 41.17±0.35 |
| NWQ†(Ours) | 2/2 | **61.65**±0.12 | **35.13**±2.16 | **56.36**±0.01 | **67.02**±0.01 | **53.55**±0.20 |

Table 1: **Comprehensive comparison**. We report top1 accuracy on the ImageNet validation set. * represents for reference from QDROP[40] and † means scaling up the calibration set to 10240 images. QDROP† results are produced by open-source codes from QDrop. We use $n = 0, m = 1$ for NWQ and $n = -1, m = 2$ for NWQ†. We follow BRECQ[27] to set the first and the last layer to 8-bit.

**Classical Calibration Set.** Following previous works, we compare the performance on a small calibration set of 1024 images. For start, we achieve 0.2%∼ 0.9% improvement even compared with strong baselines on W4A4, which are rather close to full precision accuracy. For W3A3, NWQ yields 0.83%∼ 3.26% and reduces the gap between W3A3 and W4A4. For the challenging W2A2, our method surpasses the previous state-of-the-art by a large margin of 13.37% for MobileNetV2 and 12.4% for MnasNet, pushing the limit of extremely low-bit PTQ step further.

**Scaled-up Calibration Set.** To further explore the potential of NWQ, we scale up the calibration set by $10\times$ and reproduce QDROP on the scaled-up calibration set, building a very strong baseline. Interestingly, NWQ outperforms QDROP even with one-tenth data. When equipping NWQ with the scaled-up calibration set, we observe significant improvement under all settings. Particularly, the accuracy gaps of W2A2 between NWQ and QDROP are enlarged even more as the calibration set scales up. For example, the gap of MnasNet increases from 12.4% to 19.36%, and that of MobileNetV2 increases from 13.37% to 20.24%, showing the excellent scalability of our method.

## 4.2 Ablation Study

### 4.2.1 AR

**The Effect of AR.** We study the effect of AR via comparing the experiments with layer-wise AR to the experiments without AR on W2A2. Table 2 shows that layer-wise AR improves the accuracy significantly. It's worth noting that layer-wise AR improves 5.22% for MobileNetV2 and 10.42% for MnasNet. To understand how AR works, we analyze the loss curves during training in Figure 3. First, our AR experiment consistently converges to lower loss

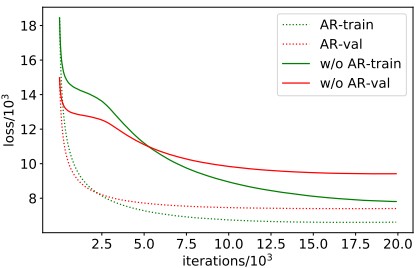

Figure 3: **AR vs. w/o AR** on MNV2 with W2A2. Quantization loss curves on train set of 1024 images and validation set of $5 \times 10^4$ images.

| AR Granularity | Res18 | MNV2 | Reg600M | Reg3.2G | Mnas |
|---|---|---|---|---|---|
| layer-wise AR | **59.14** | 26.42 | 48.49 | **62.85** | 41.17 |
| block-wise AR | 59.12 | **26.52** | **48.89** | 61.31 | **41.89** |
| stage-wise AR | 59.10 | 25.52 | 47.28 | 60.30 | 41.25 |
| w/o AR | 57.94 | 21.20 | 46.82 | 59.82 | 30.75 |

Table 2: **AR vs. w/o AR** on W2A2 with our NWQ approach. Up: Comparison of different AR granularities (*i.e.* Apply AR after each layer/block/stage). Bottom: Experiments without AR.

| AR | AA | Res18 | MNV2 | Reg600M | Reg3.2G | Mnas |
|---|---|---|---|---|---|---|
| ✗ | ✗ | 57.94 | 21.20 | 46.82 | 59.82 | 30.75 |
| ✗ | ✓ | 58.99 | 24.86 | 47.92 | 62.19 | 38.16 |
| ✓ | ✗ | 59.14 | 26.42 | 48.49 | 62.85 | 41.17 |
| ✓ | ✓ | **59.52** | **26.90** | **48.62** | **63.52** | **42.92** |

Table 3: **AR vs. AutoAugment** on W2A2. For AutoAugment, we use the open-source codes from torchvision. AA is short for AutoAugment.

| Method | RSeR | | | | | | ASoftmax |
|---|---|---|---|---|---|---|---|
| LW | 0.01 | 0.1 | 1 | 10 | 100 | 1000 | - |
| Res18 | 0.96 | 3.49 | 34.75 | **46.56** | 44.54 | 43.83 | **59.14** |
| MNV2 | 0.10 | 0.11 | 0.28 | 0.87 | **4.86** | 2.95 | **26.42** |

Table 4: **ASoftmax vs. RSeR** on W2A2 with our NWQ approach. *LW* represents the regularization loss weight for AdaRound. We simply replace ASoftmax with RSeR in our NWQ solution.

| Initialization | Bits(W/A) | ResNet18 | MobileNetV2 |
|---|---|---|---|
| random | 2/2 | 55.33 | 22.00 |
| Ours | 2/2 | **59.14** | **26.42** |

Table 5: Comparison of different initialization methods. We compare random initialization with our method described in Sec. 3.2.2.

on both the train and test datasets, resulting in higher accuracy. Second, AR accelerates convergence significantly and thus allows NWQ to reach competitive accuracy at a substantially lower computing cost. Third, even though AR does not totally eliminate over-fitting, it effectively bridges the loss gap between train and test datasets.

**The Effect of Granularity.** Table 2 studies how AR granularity affects accuracy. AR granularity represents for applying AR after each layer/block/stage, while training(reconstruction) granularity represents for optimizing each layer/block/stage sequentially and independently. We discover that lightweight networks prefer coarse-grained AR, whereas heavyweight networks prefer fine-grained AR. MobileNetV2, RegNet-600MF, and MnasNet, for example, achieve the best performance with block-wise AR, whereas ResNet18 and RegNet-3.2GF achieve the maximum accuracy with layer-wise AR.

**Comparison with Data Augmentation.** Data augmentation is a classical strategy used to relieve over-fitting due to its simplicity and effectiveness. We compare AR with strong augmentation (*e.g.* AutoAugment[8]) in Table 3. Notably, both AR and AutoAugment significantly improve accuracy, and AR contributes more than AutoAugment, particularly for sparse networks like MobileNetV2 and MnasNet. Coupling these two techniques enables us to achieve better performance.

#### 4.2.2 ASoftmax Study

**The Effect of ASoftmax.** Table 4 compares ASoftmax to Rectified Sigmoid with explicit Regularization (RSeR) which is used in AdaRound. We simply replace ASoftmax with RSeR in our NWQ and run a grid search on regularization loss weight to explore the potential of RSeR. We discover that RSeR requires a huge loss weight for NWQ, which is $10^3 \sim 10^5 \times$ larger than that of subnet-wise approaches. Even with exhausting hyper-parameter search, RSeR does not perform as well as in the subnet-wise method. ASoftmax, on the other hand, outperforms RSeR by 12.58% and 21.56% on ResNet18 and MobileNetV2, respectively, without any additional hyper-parameter tuning.

**The Effect of Initialization.** Table 5 demonstrates the effectiveness of our initialization, which keeps the quantization model close to the float-point at the start to steady the training. It is shown that our initialization method significantly outperforms random initialization.

**The Robustness of $\tau$.** We investigate the robustness of $\tau$ in Table 6. $\tau$ starts with a large number to encourage continuous relaxation and ends with a small value to favor one-hot distribution. The final $\tau^T$ determines the decaying speed of $\tau$. It can be observed that $\tau^T$ has a robustness range of $10^{-2} \sim 10^{-4}$, making it easily adaptable to a wider range of neural network architectures.

| $\tau^T$ | Res18 | MNV2 | Reg600M | Reg3.2G | Mnas |
|---|---|---|---|---|---|
| 1e-1 | 59.08 | 12.19 | 47.84 | 60.10 | 33.92 |
| 1e-2 | **59.14** | 26.42 | 48.49 | **62.85** | 41.17 |
| 1e-3 | 59.05 | **26.75** | **49.77** | 62.09 | **41.34** |
| 1e-4 | 58.98 | 24.74 | 49.42 | 62.06 | 41.32 |

Table 6: **Robustness of $\tau^T$ for ASoftmax** on W2A2. Experiments are conducted with $\tau^0 = 1.0$

| #Images | n/m | Res18 | MNV2 | Reg600M | Reg3.2G | Mnas |
|---|---|---|---|---|---|---|
| 1024 | 0/1 | 59.14 | 26.42 | 48.49 | 62.85 | 41.17 |
| 1024 | -1/2 | 59.10 | 28.73 | 49.73 | 61.68 | 41.07 |
| 10240 | 0/1 | 61.61 | 29.54 | 53.83 | 67.02 | 51.15 |
| 10240 | -1/2 | **61.65** | **35.13** | **56.36** | 67.02 | **53.55** |

Table 7: **Extensiveness of ASoftmax** on W2A2. MNV2 is short for MobileNetV2.

| Method | Bits(W/A) | ResNet18 | MobileNetV2 |
|---|---|---|---|
| QDROP-style | 2/2 | 56.47 | 5.13 |
| w/o Drop | 2/2 | 58.42 | 25.09 |
| AMixup | 2/2 | **59.12** | **26.42** |

Table 8: **QDROP-style *vs.* w/o Drop *vs.* AMixup** on W2A2 with our NWQ approach. *QDROP-style* represents for dropping methods of QDROP coupled with NWQ. *w/o Drop* represents for training without dropping or mixup operation.

| $P_s$ | Res18 | MNV2 | Reg600M | Reg3.2G | Mnas |
|---|---|---|---|---|---|
| 1 | **59.47** | 25.34 | **50.53** | **63.06** | 40.55 |
| 0.8 | 59.39 | 25.57 | 50.05 | 62.65 | 40.66 |
| 0.6 | 59.32 | 24.18 | 49.84 | 61.90 | 40.86 |
| 0.5 | 59.14 | **26.42** | 48.49 | 62.85 | **41.17** |
| 0.4 | 59.08 | 25.79 | 48.83 | 61.73 | 40.63 |
| 0.2 | 58.96 | 25.44 | 48.81 | 60.76 | 39.71 |

Table 9: **Decay Policy of AMixup** on W2A2. We use $P_e = 0$ for all of the experiments.

**The Extensiveness of ASoftmax.** As described in Sec.3.2.1, ASoftmax can easily extend the discrete parameter space from $\{0, 1\}$ to a larger integer range, offering the opportunity for a better optimum. However, extending discrete range benefits MobileNetV2 and RegNet-600MF while degrading RegNet-3.2GF on a small calibration set, as shown in Table 7 upper group. We argue that the enlarged discrete range raises the risk of over-fitting for large networks. When we scale up the calibration set, the performance no longer degrades and the performance improves even more.

### 4.2.3 AMixup Study

**The Effect of AMixup.** QDROP presents good performance in block-wise reconstruction, but fails in our network-wise scheme. As illustrated in Table 8, we achieve 1.95% improvement for ResNet18 and 19.96% for MobileNetV2 by simply removing QDROP (w/o Drop row). We hypothesize that randomly mixing activations up introduces inconsistency bias between the training and test phase, which is concealed by the improvement of flatness in block-wise reconstruction. However, as the reconstruction unit goes deeper (from block to network) and sparser, the problem becomes more pronounced, resulting in significant degeneration. Over-parameterized ResNet18 is able to compensate for the side effect, but MobileNetV2 suffers a great performance drop. Remarkably, our AMixup is able to overcome the inconsistency while taking advantage of the flatness, contributing to higher accuracy.

**The Policy of Decay.** In Table 9, we run a grid search for $P_s$ to explores the influence of different decay policies. It's worth noting that MobileNetV2 and MnasNet prefer 0.5, whereas ResNet18 and RegNet prefer 1. The basic law is very similar to that of AR granularity, which states that heavyweight networks require more regularization, implying that AMixup and AR are both capable of regularizing accuracy.

### 4.3 Efficiency

We compare the efficiency of several approaches on ResNet18. According to Figure 4, NWQ outperforms QDROP and BRECQ under various iterations. Specifically, NWQ achieve competitive results with only 10% computation cost, demonstrating the efficiency of NWQ. As the number of iterations increases, NWQ continues to improve accuracy, whereas BRECQ and QDROP reach their saturation at $\sim 1.6 \times 10^4$ iterations.

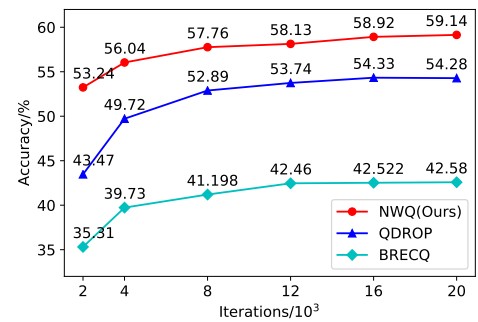

Figure 4: **Accuracy *w.r.t.* iterations**. The experiments are conducted on ResNet18 with W2A2.

## 5 Discussion and Conclusion

Simple algorithms that scale well are the core of deep learning. Prior works tend to quantize networks in layer-wise and block-wise manner. We prove that network-wise quantization is a simpler and more effective way than layer/block-wise solutions. To accomplish this, we analyze the problems introduced by the network-wise regime and propose three key designs to address them. Based on that, we construct a simple and effective end-to-end network-wise PTQ framework. Extensive experiments demonstrate that our solution is scalable, effective, and efficient. However, there is still a large accuracy gap between low-bit PTQ and full precision network, and we believe that there is still much room for further improvement for PTQ, which is a key direction for future research.

**Broder impacts.** The proposed method uses GPU resources for training, thus resulting in a certain amount of energy consumption and carbon footprints. For example, it takes about 4 GPU-hours for our method to quantize ResNet18, which would consume around 1 kWh and produce around 1 lbs of $CO_2$, accelerating climate change and global warming in some degree.

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

## A ASoftmax Initialization

First of all, We define the fractional part of $x$ as: $\{x\} = x - \lfloor x \rfloor$, and we use $\overline{\mathbf{w}}^0$ to represent initially simulated quantization of $\mathbf{w}$. According to equation (9),(10), (12) and (13), we get:

$$
\begin{aligned}
h^0(\mathbf{V}) &= \mathbb{E}_{k \sim p(\mathcal{K}_i | \mathbf{V}, \tau^t)} \left[ k \right] \\
&= \sum_{i=0}^{m-n} \mathcal{K}_i * \sigma^0(\mathbf{V})_i \\
&= \frac{1}{\sum_{i=0}^{m-n} \hat{\sigma}^0(\mathbf{V})_i} \sum_{i=0}^{m-n} \mathcal{K}_i * \hat{\sigma}^0(\mathbf{V})_i \\
&= \frac{1}{\sum_{i=0}^{m-n} \hat{\sigma}^0(\mathbf{V})_i} \sum_{i=0}^{-n} \left( \mathcal{K}_i * \hat{\sigma}^0(\mathbf{v})_i + (1 - \mathcal{K}_i) * \hat{\sigma}^0(\mathbf{v})_{m-n-i} \right) \\
&= \frac{1}{\sum_{i=0}^{m-n} \hat{\sigma}^0(\mathbf{V})_i} \sum_{i=0}^{-n} \left( \frac{\mathcal{K}_i}{\{\frac{w}{s}\} - \mathcal{K}_i} + \frac{1 - \mathcal{K}_i}{1 - \mathcal{K}_i - \{\frac{w}{s}\}} \right) \\
&= \frac{1}{\sum_{i=0}^{m-n} \hat{\sigma}^0(\mathbf{V})_i} \sum_{i=0}^{-n} \left( \frac{(1 - 2\mathcal{K}_i)\{\frac{w}{s}\}}{(\{\frac{w}{s}\} - \mathcal{K}_i)(1 - \mathcal{K}_i - \{\frac{w}{s}\})} \right) \quad (15) \\
&= \frac{\{\frac{w}{s}\}}{\sum_{i=0}^{m-n} \hat{\sigma}^0(\mathbf{V})_i} \sum_{i=0}^{-n} \left( \frac{1 - 2\mathcal{K}_i}{(\{\frac{w}{s}\} - \mathcal{K}_i)(1 - \mathcal{K}_i - \{\frac{w}{s}\})} \right)
\end{aligned}
$$

$$= \frac{\{\frac{w}{s}\}}{\sum_{i=0}^{m-n} \hat{\sigma}^0(\mathbf{V})_i} \sum_{i=0}^{-n} \left( \frac{1}{\{\frac{w}{s}\} - \mathcal{K}_i} + \frac{1}{1 - \mathcal{K}_i - \{\frac{w}{s}\}} \right)$$

$$= \frac{\{\frac{w}{s}\}}{\sum_{i=0}^{m-n} \hat{\sigma}^0(\mathbf{V})_i} \sum_{i=0}^{m-n} \hat{\sigma}^0(\mathbf{V})_i$$

$$= \{\frac{w}{s}\}$$

If we disregard the *clip* function, then

$$\begin{aligned}
\overline{\mathbf{w}}^0 &= \hat{\mathbf{w}}^0 \cdot s \\
&= (\lfloor \frac{\mathbf{w}}{s} \rfloor + h^0(\mathbf{V})) \cdot s \\
&= (\lfloor \frac{\mathbf{w}}{s} \rfloor + \{\frac{\mathbf{w}}{s}\}) \cdot s \\
&= \frac{\mathbf{w}}{s} \cdot s \\
&= \mathbf{w}
\end{aligned} \quad (16)$$

