# OpenReview forum: "Leveraging Inter-Layer Dependency for Post -Training Quantization"
_NeurIPS.cc/2022/Conference — NeurIPS 2022 Accept_

### Official Review · Reviewer_NNev · 2022-06-22

**Rating:** 4
**Confidence:** 5
**Soundness:** 3 good
**Presentation:** 3 good
**Contribution:** 2 fair

**Summary:**

### Overall:
To avoid the overfitting problem in quantized networks, the authors present several training strategies based on the previous Post-training Quantization (PTQ) methods, such as layer-wise activation fitting as proposed in BRECQ [1], Annealing Softmax (AS) to enlarge the searching space of AdaRound [2], and Annealing Mixup (AMixup) to smooth the training process of QDrop [3]. They conduct thorough ablation studies and experiments on several mainstream networks to fully verify the superiority of the proposed methods.

### Reference:
[1] Brecq: Pushing the limit of post-training quantization by block reconstruction. ICLR2021

[2] Up or down? adaptive rounding for post-training quantization. ICMLR2020

[3] Qdrop: Randomly dropping quantization for extremely low-bit post-training quantization. ICLR2022

**Questions:**

### Major concerns:
- It seems that the proposed Activation Regularization (AR) shares the same idea with BRECQ [1]. The ablation study on layer-wise/block-wise/stage-wise fitting can also be found in [1]. Why do the authors list AR as the contribution of this paper? What is the definition of "inter-layer dependency"? What is the relationship between "inter-layer dependency" and the overfitting problem?
- The proposed methods heavily rely on hyper-parameter tuning, as reported in Section 4.2. I do notice that the authors conduct ablation studies in most settings. However, I doubt the search space can be too large. Did the authors apply the same setting (i.e., same AR, $\tau^T$, $n/m$, $P_s$) to all experiments reported in Table 1? Besides, did the authors conduct ablation studies using the validation set and also report the final results on the validation dataset in Table 1?
- It is interesting to see that the $59.14, 26.42, 48.49, 62.85, 41.17$ widely exist in Tables 2-8. How to determine the best setting of AR, $\tau^T$, $n/m$, and $P_s$ simultaneously?

### Minor concerns:
- It seems that the selection of $n/m$ is quite sensitive to the size of the calibration dataset. Besides, the authors may further explain the reason for using $0/1$ instead of $-1/2$.
- I am not sure that it is reasonable to use 10240 calibration images in the circumstances of post-training quantization. It would be better to include a simple baseline utilizing naive Knowledge Distillation (KD) based on 10240 images to fine-tune a quantized network.
- Table 5: I am curious why ASoftmax with an integer range of $\\{-1,0,+1,+2\\}$ outperforms RSeR [3] with $\\{-1, 0, +1\\}$ in 2-bit. Note that 2-bit weights only contain $\\{-2, -1, 0, +1\\}$. The results listed in Table 5 seems to indicate that ASoftmax turns some $-2$ and $-1$ into $0$ and $+1$, is that correct? It would be better to include $n/m=-1/+1$ to make a fair comparison with RSeR.
- The QDrop [2] results reported in Table 4 seem inconsistent with Table 1 and Res18 AMixup with 59.12% Top-1 Acc. is slightly lower than Table 2.
- The ablation study on the **Initialization** of ASoftmax seems missing.

### Reference:
[1] Brecq: Pushing the limit of post-training quantization by block reconstruction. ICLR2021

[2] Qdrop: Randomly dropping quantization for extremely low-bit post-training quantization. ICLR2022

[3] Up or down? adaptive rounding for post-training quantization. ICMLR2020

**Limitations:**

### Comments on limitations:
As claimed in the **Abstract**, "This process pays little attention to the dependency across the sub-nets, hence is less optimal." However, the main idea has been rarely discussed in the current manuscript, which makes the submission serve more like a technical report (i.e., the descriptions and effect of different improvements). The authors are encouraged to solve the major concerns in the final version.

**Strengths And Weaknesses:**

### Strengths:
- The experimental results seem strong. The authors conduct extensive experiments on ImageNet with different bitwidth. Especially, the ablation studies are quite clear, which should be encouraged.
- The technique part of this paper is well written. I personally think this paper is simple and easy to follow.
- "Annealing Softmax" seems new in the community of model compression.

### Weaknesses:
- Please refer to *Questions* and *Limitations*. The main concern is the novelty of this paper.

---

> ### Author Response · Authors · 2022-07-30
> **Response to Reviewer NNev Part II**
>
> * **Q7**: I am not sure that it is reasonable to use 10240 calibration images in the circumstances of post-training quantization. It would be better to include a simple baseline utilizing naive Knowledge Distillation (KD) based on 10240 images to fine-tune a quantized network.
>
>   **A7**: It should be noted that we first achieve SOTA results on classical 1024 images following previous works in Table 1, then we scale up the calibration set to 10240 images to further demonstrate the scalability of our method. We also run the QDROP experiments with 10240 images for a fair comparison. We have conducted naive KD with 10240 images but the performance is too low to be a baseline.
> &nbsp;
> * **Q8**: Table 5: I am curious why ASoftmax with an integer range of {−1,0,+1,+2} outperforms RSeR [3] with {−1,0,+1} in 2-bit. ...
>
>   **A8**: We suppose that the reviewer may have misunderstood Table 5 (Table 4 in our revised version), where the integer ranges of all experiment are {0, 1}. In fact, RSeR is designed for binary discretization thus only supports {0,1}. Extended ASoftmax does have the ability to turn $h(v)$ to arbitrary integer in {-2, -1, 0, 1}. The comparison between RSeR and ASoftmax is fair since we have already used 0/1 for n/m in Table 5.
> &nbsp;
> * **Q9**: The QDrop [2] results reported in Table 4 seem inconsistent with Table 1 and Res18 AMixup with 59.12% Top-1 Acc. is slightly lower than Table 2.
>
>   **A9**: We are sorry for the confusing expression. QDROP row in Table 4(Table 8 in our revised version) means applying QDROP-style dropping policy in our NWQ training, rather than QDROP original results. We have corrected this table in our revised manuscript.
> &nbsp;
> * **Q10**: The ablation study on the Initialization of ASoftmax seems missing.
>
>   **A10**: Thank you for your thoughtful advice. We have included related experiments in our revised version (see Table 5 on page 8).
> &nbsp;
> * **Q11**: As claimed in the Abstract, ... However, the main idea has been rarely discussed in the current manuscript, ...
>
>   **A11**: Thank you for your thoughtful and constructive feedback. We have added more discussion about our main idea to introduction in our revised manuscript(L38-L54).
>
> Reference:
>
> [1] Brecq: Pushing the limit of post-training quantization by block reconstruction. ICLR2021
>
> [2] Up or down? adaptive rounding for post-training quantization. ICMLR2020
>
> [3] Qdrop: Randomly dropping quantization for extremely low-bit post-training quantization. ICLR2022

---

> ### Author Response · Authors · 2022-07-30
> **Response to Reviewer NNev Part I**
>
> Thank you for your thoughtful comments on our work. We address your main concerns in above "General Response to All Reviewers", and we provide additional details on specific comments below.
>
>
> * **Q1**: It seems that the proposed Activation Regularization (AR) shares the same idea with BRECQ [1]. The ablation study on layer-wise/block-wise/stage-wise fitting can also be found in [1]. Why do the authors list AR as the contribution of this paper?
>
>   **A1**: We respectfully disagree with you that AR shares the same idea with BRECQ. Their functionalities are different. BRECQ's activation losses are quantization objective functions while AR works as a regularizer. BRECQ points out that *"optimizing the whole networks over 1024 calibration data samples leads to over-fitting easily"*[1] and we have observed the same phenomenon. We also found that the quantized output is very close to the float results but the intermediate activations drift far from the float counterparts, so we propose to apply regularization on intermediate activations. Experiments (Figure 3&Table 3) demonstrate that AR effectively alleviates over-fitting.
> &nbsp;
> * **Q2**: What is the definition of "inter-layer dependency"?
>
>   **A2**: We have defined "inter-layer dependency" in our introduction: the capability that *quantization errors of all layers in a network are able to compensate for each other, resulting in a reduced quantization error*, which is inspired by AdaRound[2].
> &nbsp;
> * **Q3**: What is the relationship between "inter-layer dependency" and the overfitting problem?
>
>   **A3**: "Inter-layer dependency" motivates us to jointly optimize all layers which greatly enlarges the optimization space, and larger optimization space over limited calibration data results in higher risk of over-fitting.
> &nbsp;
> * **Q4**: The proposed methods heavily rely on hyper-parameter tuning, as reported in Section 4.2. ...
>
>   **A4**:  Actually our method is robust to hyper-parameters. The ablation studies demonstrate that these hyper-parameters have rather wide robust ranges, the gain of  hyper-parameter tuning is marginal compared to our overall improvement.
>
>      We use the default setting across our entire paper, including SOTA results in Table 1.
>
>      All of our results are reported on ImageNet1k validation set.  Please refer to "General Response to All Reviewers" about hyper-parameters for more details.
> &nbsp;
> * **Q5**: It is interesting to see that the 59.14, 26.42, 48.49, 62.85, 41.17 widely exist in Tables 2-8. How to determine the best setting of AR, $\tau^T,n/m$,and $P_s$ simultaneously?
>
>   **A5**: 59.14,26.42,...,41.17 widely exist in Tables 2-8 because all these experiments use the same setting (our default setting). Please refer to "General Response to All Reviewers" about hyper-parameters for best setting.
> &nbsp;
> * **Q6**: It seems that the selection of $n/m$ is quite sensitive to the size of the calibration dataset. Besides, the authors may further explain the reason for using $0/1$ instead of $-1/2$.
>
>   **A6**: The best choice of $n/m$ does have relation to data size. However, the default 0/1 has already achieved excellent results across various data size, and the extending of $n/m$ is actually extra bonus which requires more data to avoid over-fitting. Moreover, there is a basic law to guide the selection, please refer to "General Response to All Reviewers" about hyper-parameters best setting for more details.
>
>     We choose 0/1 by default because previous works' learning space is {0, 1} and we want the comparison easy to clarify. We only use -1/2 for scaled-up calibration dataset to demonstrate the scalability of our method.

---

### Official Review · Reviewer_N5CW · 2022-07-10

**Rating:** 7
**Confidence:** 3
**Soundness:** 3 good
**Presentation:** 3 good
**Contribution:** 3 good

**Summary:**

This paper proposes a post-training quantization (PTQ) technique that aims to search for a network-level quantization policy, compared to the previous works that only considers quantization policies on a layer-wise level. To tackle the challenges of solving a larger scale combinatorial optimization problem, and the issue of overfitting, the paper proposes two approaches: Activation Regularization, and Annealing Softmax and Annealing Mixup. Experiments show the proposed approach achieved higher performance especially when quantizing networks to extremely low bits.

**Questions:**

When explaining the effect of AR, the author mentioned two roles: 1/ preventing overfitting and 2/ preventing the quantized values deviating from the floating point counter part. I can understand the effect of 2/, but 1/ is not clear to me. Are there more evidence supporting why overfitting is an issue?

**Limitations:**

1/ Seems that the performance gain is not consistent. The performance gain of the proposed method is significant for some configurations, e.g., MobileNet with 2 & 3 bit quantization. However, the gain is quite small for others, such as MobileNetV2 with 4-bit quantization. Why is that? Is it simply because MobileNetV2 2&3 bit configurations do not have a strong baseline?

2/ The proposed method seem to contain too many hyper-parameters, from the granularity of AR, to initialization scheme, to annealing schedules, etc. How difficult is it to find a good configuration for a new model and new target bits?



**Strengths And Weaknesses:**

1/ The insight of this paper is straight-forward but very sensible -- quantization errors can cancel each other, and optimizing with a higher granularity can achieve better performance. More importantly, based on the insight, the authors proposes effective approaches that achieved this.

2/ The paper is clearly written. The derivation and introduction of the techniques are explained very clearly.

3/ Nice ablation study shedding lights on the hyper-parameter sensitivity of the proposed method.

---

> ### Author Response · Authors · 2022-07-30
> **Response to Reviewer N5CW**
>
> We thank Reviewer N5CW for positive comments and for providing thoughtful feedback on our work. Below is the detailed response to each question, hope you can find them helpful.
>
> * **Q1**: Are there more evidence supporting why overfitting is an issue?
>
>   **A1**: Yes. First, BRECQ admitted that they failed to build a network-wise solution because *"optimizing the whole networks over 1024 calibration data samples leads to over-fitting easily"*[1]. In addition, it can be observed in Figure 3(L236-L250 in our revised manuscript) that the training loss is rather lower than test loss and the loss gap is narrowed after applying AR. Intuitively, given only 1024 images, end-to-end quantization enlarges optimization space thus leads to over-fitting issue.
> &nbsp;
> * **Q2**: Seems that the performance gain is not consistent. The performance gain of the proposed method is significant for some configurations, e.g., MobileNet with 2 & 3 bit quantization. However, the gain is quite small for others, such as MobileNetV2 with 4-bit quantization. Why is that? Is it simply because MobileNetV2 2&3 bit configurations do not have a strong baseline?
>
>   **A2**: We think there are two reasons. The first is, as Reviewer N5CW states, that the strong baselines of 4 bits are very close to full precision results. The second is that lower bits make the independency hypothesises of AdaRound and BRECQ less accurate, as BRECQ points out that AdaRound *"cannot further quantize the weights into INT2 because the cross-layer dependency in the Hessian matrix cannot be ignored when the perturbation on weight is not small enough."*[1] This makes the benefits of NWQ more significant on lower bits.
> &nbsp;
> * **Q3**: The proposed method seem to contain too many hyper-parameters, from the granularity of AR, to initialization scheme, to annealing schedules, etc. How difficult is it to find a good configuration for a new model and new target bits?
>
>   **A3**: For new model and new target bits, we recommend our default setting which practically generalizes well across various architectures and bit widths. Please refer to our "General Response to All Reviewers" about hyper-parameters robustness and best-tuning for more details.
>
> Reference:
>
> [1] Brecq: Pushing the limit of post-training quantization by block reconstruction. ICLR2021
>
> [2] Up or down? adaptive rounding for post-training quantization. ICMLR2020
>
> [3] Qdrop: Randomly dropping quantization for extremely low-bit post-training quantization. ICLR2022

---

### Official Review · Reviewer_GPuz · 2022-07-11

**Rating:** 4
**Confidence:** 3
**Soundness:** 3 good
**Presentation:** 3 good
**Contribution:** 2 fair

**Summary:**

Based on the inter-layer dependency of quantization, this paper proposes a Network-Wise Quantization (NWQ) approach. NWQ arises overfitting and discrete optimization problems.
In this paper, activation regularization is introduced to handle the overfitting issue. To solve the combinatorial optimization problem,  the authors use Annealing Softmax and Anealing Mixup.
The experiments show that NWQ can achive significant improvements over previous works.


**Questions:**

According to paper of AdaRound and Brecq, the inferred layer-wise frobenius loss is based on the hypothesis that the Hessian matrix H is diagonal, which means that the optimization of each layer are independent. Moreover, Brecq has validated the finer granularity brought better quantization quality, which contradict with the idea of inter-layer dependency in this paper, can you explain more on that part?

**Limitations:**

Limited technical contribution. The proposed methods are largely based on previous papers. For the new techniques, no theoretical analysis is provided. For example, no theoretical analysis of ASoftmax regarding the better performance compared with RSeR.

**Strengths And Weaknesses:**

Strengths:
PTQ is widely studied, and the methods proposed in this paper outperfome previous SOTA by a large margin. The paper is easy to follow and the authors provides several ablation study to show the effectiveness of proposed techniques.

Weakness:
The paper shows some originality but is not novel enough. For example, Annealing Softmax and Annealing Mixup are largely based on AdaRound[32] and QDROP[41]. The authors clearly state the difference and demonstrate the effectiveness of modifications, but previous methods play a more important role in the discrete optimization problem.  Besides, the motivation and theoretical analysis of substituting RSeR with ASoftmax are absent.

From finger 3, we can learn the efficiency of NWQ. However, the cost of each iteration is not clear. Does it require less or more computation/memory than previous methods in each iteration.

---

> ### Author Response · Authors · 2022-07-30
> **Response to Reviewer GPuz**
>
> Thank you for your time and efforts on our work. We address your main concerns in above "General Response to All Reviewers", and we provide additional details on specific comments below.
>
> * **Q1**: According to paper of AdaRound and Brecq, the inferred layer-wise frobenius loss is based on the hypothesis that the Hessian matrix H is diagonal, which means that the optimization of each layer are independent.
>
>   **A1**: It should be noted that both the layer-diagonal hypothesis of AdaRound[2] and block-diagonal hypothesis of BRECQ[1] are not accurate. AdaRound assumes that layers are independent, while BRECQ[1] points out that AdaRound *"cannot further quantize the weights into INT2 because the cross-layer dependency in the Hessian matrix cannot be ignored when the perturbation on weight is not small enough"*[1], so BRECQ[1] extends Hessian from the layer-diagonal to the block-diagonal based on their block independency hypothesis. However, BRECQ's hypothesis is still not accurate because they have ignored the dependency across blocks. We have proved that fully leveraging the inter-layer dependency across entire networks leads to higher performance.
> &nbsp;
> * **Q2**: Moreover, Brecq has validated the finer granularity brought better quantization quality, which contradict with the idea of inter-layer dependency in this paper, can you explain more on that part?
>
>   **A2**: In fact, BRECQ[1] concludes that block-wise reconstruction
>   is *"a better bias-variance trade-off choice"*[1] compared to finer layer-wise and coarser stage-wise/network-wise reconstruction. BRECQ utilizes inter-layer dependency within each block via extending layer-wise method to block-wise reconstruction and achieves better performance, which just agrees with our idea actually. However, BRECQ failed to build a network-wise solution because they found that *"optimizing the whole networks over 1024 calibration data samples leads to over-fitting easily"*[1] and they didn't resolve this problem. Our work significantly alleviates over-fitting problem thus achieves better performance.
> &nbsp;
> * **Q3**: From finger 3, we can learn the efficiency of NWQ. However, the cost of each iteration is not clear. Does it require less or more computation/memory than previous methods in each iteration.
>
>   **A3**: The cost of each iteration for all the methods are the same. We use the same batch size and GPU resources for all the methods for a fair comparison. For a network with 10 blocks, BRECQ/QDROP will optimize the 10 blocks sequentially and each block runs for T iterations. NWQ directly optimize the entire network for T iterations. Theoretically, the computation costs of each iteration for all methods are the same. The efficiency of NWQ results from faster convergence thus is able to achieve competitive results with much less iterations, rather than unfair comparison.
> &nbsp;
> * **Q4**: Limited technical contribution. The proposed methods are largely based on previous papers. For the new techniques, no theoretical analysis is provided. For example, no theoretical analysis of ASoftmax regarding the better performance compared with RSeR.
>
>   **A4**: Please refer to "General Response to All Reviewers" above where we have summarized our novelty and technical contributions. For the analysis of RSeR, please refer to Sec 3.2(L152-L161) in our revised manuscript. We hope it can address your concern.
>
> Reference:
>
> [1] Brecq: Pushing the limit of post-training quantization by block reconstruction. ICLR2021
>
> [2] Up or down? adaptive rounding for post-training quantization. ICMLR2020
>
> [3] Qdrop: Randomly dropping quantization for extremely low-bit post-training quantization. ICLR2022

---

> > ### Comment · Reviewer_GPuz · 2022-08-09
> > **Response to authors**
> >
> > Thank you for the detailed response. Although I still consider this paper somewhat incremental, your rebuttal regarding the novelty has clarified the contributions of this paper more clearly. Also, you provide more analysis of RSeR, especially the drawbacks compared with Annealing Softmax.
> >
> > One thing still bothering me is the lacking mathematical support of the idea of inter-layer dependency in your paper. If you look at the derivation of loss in either Brecq or Adaround, the granularity of the block is consistent with the "block" granularity. The independency or the diagonal Hessian matrix is pre-requisite for the derivation of the frobenius loss.
> >
> > Given all the above reasons, I decided to raise the score to 4.

---

> > > ### Author Response · Authors · 2022-08-10
> > > **Response to Reviewer GPuz about mathematical support.**
> > >
> > > Thanks for your promotion and your thoughtful advice.
> > >
> > > We didn't include the mathematical support because we thought inter-layer dependency is one of the natures of CNNs. However, we will include the mathematical support in our final version to make it more theoretically convincing .
> > >
> > > In fact, the mathematical support of inter-weight dependency has been provided in AdaRound[2] by expanding objective function $\Delta w^\intercal H^{(w)} \Delta w$ to $\Delta w_1^2 + \Delta w_2^2 + \Delta w_1\Delta w_2$ with a toy example for one-layer network. If we consider that $w_1$ and $w_2$ come from different layers,  the objective function can be expanded as
> > > $$
> > > \Delta w^\intercal H^{(w)} \Delta w =
> > > \left[\matrix{
> > > \Delta  w_1^\intercal& \Delta  w_2^\intercal
> > > }\right]
> > > \left[\matrix{
> > > H^{(w)}_\{1,1\} & H^{(w)}_\{1,2\} \\\\
> > > H^{(w)}_\{2,1\} & H^{(w)}_\{2,2\}
> > > }\right]
> > > \left[\matrix{
> > >   \Delta  w_1 \\\\
> > >   \Delta  w_2
> > > }\right]
> > > = \Delta w_1^\intercal H_\{1,1\}^\{(w)\} \Delta w_1 + \Delta w_2^\intercal H_\{2,2\}^\{(w)\} \Delta w_2 + \Delta w_1^\intercal H_\{1,2\}^\{(w)\} \Delta w_2 + \Delta w_2^\intercal H_\{2,1\}^\{(w)\} \Delta w_1
> > > $$
> > > where $ \Delta w_1^\intercal H_\{1,1\}^\{(w)\} \Delta w_1 + \Delta w_2^\intercal H_\{2,2\}^\{(w)\} \Delta w_2$ are the intra-layer items while
> > > $ \Delta w_1^\intercal H_\{1,2\}^\{(w)\} \Delta w_2 + \Delta w_2^\intercal H_\{2,1\}^\{(w)\} \Delta w_1$ are the inter-layer items. The latter two items demonstrate that the interact between the two layers are able to reduce quantization error.  We will include the improved mathematical support   in our final version.
> > >
> > > Our loss design has nothing to do with the independency assumption. Based on the core idea to minimize the output quantization error end-to-end, we choose frobenius loss, which actually is the same thing with L2 loss, to distill the quantized model.  Frobenius is just one of the choices of Knowledge Distillation(KD) loss. Besides, we use the term "frobenius" instead of L2 because the output and activations are multiple dimension matrices and we want our expression more rigorous. The diagonal Hessian is not a pre-requisite for our loss design.
> > >
> > > Reference:
> > >
> > > [1] Brecq: Pushing the limit of post-training quantization by block reconstruction. ICLR2021
> > >
> > > [2] Up or down? adaptive rounding for post-training quantization. ICMLR2020
> > >
> > > [3] Qdrop: Randomly dropping quantization for extremely low-bit post-training quantization. ICLR2022

---

> ### Author Response · Authors · 2022-08-09
> **Kindly remind: rebuttal discussion is ending**
>
> Since the rebuttal discussion is about to end soon, please let us know whether our replies have addressed your questions. Please contact us if any further clarifications required or any other further concerns.

---

### Official Review · Reviewer_kWyx · 2022-07-11

**Rating:** 6
**Confidence:** 5
**Soundness:** 4 excellent
**Presentation:** 3 good
**Contribution:** 3 good

**Summary:**

The paper studies post-training quantization, and propose to leverage inter-layer dependency to improve the post-training quantized network. The authors introduce a number of approaches: AR, ASoftmax, improved initialization and AMixup, and verify the effectiveness of each part accordingly. The experimental results demonstrate superiority over a number of previous strong PTQ baselines across various network architectures and quantization configurations. In summary, despite the paper is methodologically incremental compared with past literature, the experiments are thorough and solid.

**Questions:**

- While the paper is entitled with "leveraging layer-wise dependency for PTQ", layer-wise training still achieves the best on 2 out of 5 architectures according to Table 2. I wonder what is the effect to prolong the training iterations? Will it allow the network adjust itself to better leverage the inter-layer dependency, and alleviate the convergence issue (all variables are discretized instead of being left in continuity)?

- The empirical effect of initialization (proposed in Section 3.2.2.) is missing?

- According to Table 7, ASoftmax does not bring consistent improvement with 1024 images, and may even suffer from overfitting (L254). Is there a systematic way to determine the optimization space with the training size? Additionally, have you tried larger discrete optimization space given 10240 images?

- Typo: in L159, $\hat$ is placed improperly for $s\cdot w^0$.

- Equations should end up with "," or ".".

**Ethics Review Area:**

["I don’t know"]

**Limitations:**

The authors improves the performance of post-training quantized networks under low-bit configurations. Despite there is still a large gap with the full-precision model, the empirical improvement in this paper is still an important step for futural PTQ research.

**Strengths And Weaknesses:**



Strengths:


- The major strength for this paper should be the empirical results, which outperforms a number of strong baselines and pushes the limit of PTQ performance on low-bit quantized networks. The ablations are also thorough and complete.

- The writing is clear and easy to follow.


Weakness:

- The paper is methodologically incremental as a large portion is built upon past efforts (BRECQ and QDROP). Activation regularization resembles knowledge distillation, a widely adopted approach in PTQ approaches utilizing calibration dataset. Asoftmax follows AdaRound and extends the range of discrete space in $h(v)$. Annealing mixup considers to add annealing process into the mixture of quantized and full-precision activations.

---

> ### Author Response · Authors · 2022-07-30
> **Response to Reviewer kWyx**
>
> We thank Reviewer kWyx for positive comments and helpful feedback on our work.
> We address some of Reviewer kWyx's concerns in above "General Response to All Reviewers" and we response to specific questions below.
>
> * **Q1**: While the paper is entitled with "leveraging layer-wise dependency for PTQ", ...
>
>   **A1**: The reviewer might have misunderstood Table 2. Experiments in Table 2 are all conducted with our network-wise training, but with different AR granularity. Prolonging the training iterations might allow light networks to better leverage the dependency, but raise the risk of over-fitting for large networks. The convergence problem emerges if we use RSeR instead of ASoftmax. Prolonging the training iterations might slightly alleviate the convergence issue but is very inefficient and ineffective according to our experiments.
> &nbsp;
> * **Q2**: The empirical effect of initialization (proposed in Section 3.2.2.) is missing?
>
>   **A2**: Thanks for your helpful advice, we have included the related experiments in our revised manuscript (Table 5 on page 8 in our revised manuscript). According to Table 5, our initialization significantly outperformance random initialization.
> &nbsp;
> * **Q3**: According to Table 7, ASoftmax does not bring consistent improvement with 1024 images, and may even suffer from overfitting (L254). Is there a systematic way to determine the optimization space with the training size? Additionally, have you tried larger discrete optimization space given 10240 images?
>
>   **A3**: ASoftmax brings consistent improvement compared to RSeR, as show in Table 5(Table 4 in our revised version). Table 7 further explores the potential of ASoftmax via extending the learnable range to a wider range. Let B denote the learnable range and P denote the model's weights, then the optimization space can be quantified as $|B|^{|P|}$. The basic rule is that given 1024 images, extending B for networks whose $|P|$ larger than 10M may raise the risk of over-fitting. Given 10240 images, extending B to wider range is very safe according to our experiments. We have extended B to {-2, -1, 0, 1, 2, 3} but the improvement is marginal.
> &nbsp;
> * **Q4**: Typo: in L159, \hat is placed improperly for $s\cdot w^0$. Equations should end up with "," or ".".
>
>   **A4**: Thank you for your thoughtful advice. We have updated the related part in our revised version.
>
> Reference:
>
> [1] Brecq: Pushing the limit of post-training quantization by block reconstruction. ICLR2021
>
> [2] Up or down? adaptive rounding for post-training quantization. ICMLR2020
>
> [3] Qdrop: Randomly dropping quantization for extremely low-bit post-training quantization. ICLR2022

---

> > ### Comment · Reviewer_kWyx · 2022-08-08
> > **Response to authors**
> >
> > Thanks for the authors' response and additional experiments. It would be better if the revision are colored so that we can easily figure out the difference.
> >
> >
> > The major advantage of this paper lie in the strong empirical results. Methodologically, while there are some differences with previous works as stated in the general response (e.g., end-to-end PTQ framework, activation regularization, annealing softmax, and annealing mixup e.t.c.), it does not  convince me to be "novel" enough. The good thing is that authors provide detailed ablations to verify most proposed components.
> >
> >
> > Q1, Q2 are resolved. Thanks for the explanation. It's a bit misleading between the concept of objective granularity and end-to-end/block-wise training. Hope more clarifications can be provided in the revision. For Q3, the authors mention "over-fitting" multiple times as a challenge for PTQ. However, it is still not clear how to observe "over-fitting" as well as its potential drawback empirically.

---

> > > ### Author Response · Authors · 2022-08-09
> > > **Response to Reviewer kWyx about Over-fitting**
> > >
> > > Thanks for your thoughtful advice, we have colored the revised content and provided more clarifications about the difference between AR granularity and training(reconstruction) granularity (see Table 2 & L248-L250).
> > >
> > > The over-fitting phenomenon can be observed in Figure 3(right of L235-L245 in our revised manuscript) where the training loss is rather lower than test loss without AR,  and the loss gap is effectively narrowed when applying AR.  Intuitively, given only 1024 images, end-to-end quantization enlarges optimization space thus leads to over-fitting issue. The drawback of over-fitting is that it prevent previous works from building an end-to-end PTQ as BRECQ admitted that they failed to build an end-to-end solution because "optimizing the whole networks over 1024 calibration data samples leads to over-fitting easily"[1].
> > >
> > >
> > > Reference:
> > >
> > > [1] Brecq: Pushing the limit of post-training quantization by block reconstruction. ICLR2021

---

### Author Response · Authors · 2022-07-30
**Response to All Reviewers**

We would like to thank all the reviewers for their thoughtful and constructive comments on our manuscript. We are encouraged to see that all reviewers acknowledge the SOTA results brought by our simple yet effective method.

We will reply to your comments one by one, but first of all, we'd like to make a general response to clarify some common questions:

* Regarding the **novelty**, we are the **first to successfully build an end-to-end PTQ framework** which we believe will be the new paradigm for future research on PTQ. Former researches such as BRECQ failed to train an end-to-end PTQ and turned to a workaround of block-by-block method via hypothesizing that the quantization variables of different blocks are independent. However, our work demonstrates that the dependency is not only within blocks but also across blocks and should never be neglected. By leveraging the inter-layer dependency across the entire network, our end-to-end NWQ pushes the limit of PTQ performance by a large margin. Meanwhile, the training of PTQ will be much more convenient in an end-to-end manner, without a tedious layer-by-layer/block-by-block process like previous works. We emphasize that we have very different motivation, goal, methodology and technical results from previous works:
    * **motivation**：Our motivation is to fully leverage the inter-layer dependency across the entire network to achieve better performance. For example, to avoid full Hessian computation, AdaRound[2] neglects the inter-layer dependency for a layer-diagonal approximated Hessian and BRECQ[1] restricts the dependency within each block for a block-diagonal approximated Hessian, both of which are not accurate enough thus sub-optimal.
    * **goal**: Our goal is to build a network-wise, end-to-end PTQ framework, which has never been done before to the best of our knowledge. Previous works on PTQ are built either layer-wisely or block-wisely, rather than network-wisely (*i.e.* end-to-end)
    * **methodology**: We are not trivially combining previous techniques which fail in end-to-end PTQ actually (explained below). Instead, we first point out the two major challenges (over-fitting and discrete optimization problem) brought by end-to-end PTQ scheme and then propose three core techniques to address them:
      * **Activation Regularization** works as a regularizer in our end-to-end method, while AdaRound[2] and BRECQ[1] utilize the activation losses as their quantization objective function.
      * **Annealing Softmax** provides a mechanism to guarantee the discretization automatically via annealing temperature for end-to-end PTQ, while RSeR (proposed by AdaRound[1]) encounters convergence problem when used in end-to-end PTQ.
      * **Annealing Mixup** resolves the train-test inconsistency problem introduced by QDROP[3]. This inconsistency is neglected by QDROP and will result in dramatical performance degradation in end-to-end PTQ.
    * **technical results**: We have achieved consistent improvement across 5 architectures and 3 bit widths over previous SOTA. For MobileNetV2 with 2 bits, we surpass previous SOTA by 20.24%, pushing the limit of extremely low-bit PTQ.

* Regarding the **hyper-parameters**, our method is robust to all the 4 hyper-parameters. We achieve consistent improvement with the default setting (see Table 1).
    * **robustness**: We are **NOT** heavily tuning these hyper-parameters for SOTA results. We use the **default setting** (described in "Implementation Details"Sec.4) **across the entire paper** if not explicitly specified. According to our extensive experiments, our default setting generalizes well across various architectures and bit widths.
    * **best setting**: Though the above hyper-parameters are robust, you can still improve the results slightly by tuning. According to the ablation study, the basic rule to tune these hyper-parameters is:
      * larger networks prefer finer AR granularity, smaller integer range(n/m=0/1), and larger $P_s$
      * larger calibration datasets prefer larger integer range(n/m=-1/2)
      * $\tau^T$ is robust to networks and data size thus needs no further tuning.

We have updated our manuscript, the changes we made include:
* Introduction(L38-L54): We add more discussion about our insights.
* Introduction(Figure 1, top of page 2): We include a figure to illustrate the key differences between our method and previous works.
* Approach(Sec 3.2, L152-L161): We add the discussion about the motivation of substituting RSeR.
* Ablation Study(Table 5 on page 8): We add the ablation study *w.r.t.* our initialization method.
* Ablation Study(Caption of Table 2, 4 & 8): Make the experiment settings more clear.

Reference:

[1] Brecq: Pushing the limit of post-training quantization by block reconstruction. ICLR2021

[2] Up or down? adaptive rounding for post-training quantization. ICMLR2020

[3] Qdrop: Randomly dropping quantization for extremely low-bit post-training quantization. ICLR2022

---

### Meta-Review · Area_Chair_fk9o · 2022-08-29

**Recommendation:** Accept
**Confidence:** Certain

**Metareview:**

This paper studies post-training quantization by proposing Network-Wise Quantization (NWQ) an end-to-end quantization approach that takes into account relationships between layers rather than treating layers independently. Using this approach, the paper demonstrates compelling empirical gains across a number of architectures and compression factors. Reviewers recognized the practical success of the approach as demonstrated by these empirical results and praised the clarity of the manuscript. However, there were concerns regarding the novelty of the approach and whether the proposed method is simply a composition of previous methods. While I understand these concerns, I think there is a significant delta between this work and previous approaches, especially when taking into account the markedly improved performance and the challenges of determining how to apply these lines of thinking to end-to-end training. The authors also expanded their discussion of these works in their updated manuscript, clarifying the differences. There were also concerns regarding the hyperparameter tuning, but the authors clarified in their response that the large majority of experiments used a constant set of hyperparameters, suggesting that these results are not simply the effect of tuning. Altogether, I think this paper makes an impactful contribution and will be a valuable addition to the conference.

**Award:**

No

---

### Decision · Program_Chairs · 2022-09-14

Accept